# Writing in the air: Facilitative effects of finger writing in older adults

Yoshihiro Itaguchi[1]*, Chiharu Yamada[2☯], Kazuyoshi Fukuzawa[2☯]

**1** Department of Computer Science, Shizuoka University, Hamamatsu, Japan, **2** Department of Psychology, Waseda University, Tokyo, Japan

☯ These authors contributed equally to this work.
* itaguchi-y@inf.shizuoka.ac.jp

## Abstract

*Kūsho*, which refers to a behavior in which one moves the index finger as a substitute for a pen in the air or on a surface, mostly used when trying to recall the shape of a written character or the spelling of a word, has been known to facilitate cognitive task performance among kanji writing-system users. This study investigates whether the facilitative effect of kūsho, the existence of which has been exclusively confirmed in younger adults, is present in old age. Moreover, to further understand the interaction between finger movement and cognitive processing, we analyzed the correlation between the kūsho effect and factors such as age, mini-mental state examination (MMSE) score, and number of years of education. The kūsho effect was assessed by a task where participants mentally assembled a set of kanji subparts to form an actual character. The results showed a significant facilitative effect of kūsho and a strong negative correlation between kūsho effect and education. This study confirms the benefits of finger movement for solving cognitive tasks involving visual processing of written language among older adults and suggests the kūsho effect may be mediated by education.

## Introduction

Aging is known to influence various areas of cognitive processing such as working memory [1], cognitive flexibility [2], inhibitory function [3], semantic memory [4], and certain aspects of language [5]. The present study aimed to investigate how aging influences the facilitative effect of finger movements known as *kūsho* on cognitive task performance among older Japanese adults. Kūsho is a behavior in which one moves the index finger as a substitute for a pen in the air or on a surface, mostly used when trying to recall the shape of a written character or the spelling of an English word; this behavior is mainly observed among people who use a Chinese character writing system (kanji) such as Chinese and Japanese people [6–8]. Based on a sample of Japanese young adults, previous studies found that visual feedback from kūsho behavior improved cognitive task performance for recalling the shapes of characters [9, 10]. Kūsho behavior has also shown some benefits for second-language learners of Japanese [11, 12]. While kūsho behavior is widely observed among Japanese people regardless of age, no

**Data Availability Statement:** All relevant data are within the manuscript and its Supporting Information files.

**Funding:** Japan Society for the Promotion of Science http://dx.doi.org/10.13039/501100001691

KAKENHI 18K13372 Dr. Yoshihiro Itaguchi Japan Society for the Promotion of Science http://dx.doi.org/10.13039/501100001691 KAKENHI 18K03188 Prof. Kazuyoshi Fukuzawa The funders had no role in study design, data collection and analysis, decision to publish, or preparation of the manuscript.

**Competing interests:** The authors have declared that no competing interests exist.

previous study has specifically confirmed the effect of kūsho behavior on cognitive processing among older adults. The present study, therefore, aimed to investigate whether the facilitative effect of kūsho behavior is observed among kanji-culture older adults as it is among younger adults, regardless of age-related changes in the sensorimotor system and cognitive functions.

We used a kanji construction task to investigate the existence of the facilitative effect of kūsho behavior in older adults. In this task, participants try to assemble three presented kanji subparts to form an actual existing kanji character [6, 9, 10]. Kanji characters usually consist of several subparts and therefore can be broken down into several units. Kūsho behavior is frequently observed among kanji-culture individuals [6, 7, 13–15]. Even in experimental settings, the spontaneous use of this behavior is widely seen among Japanese university students [6, 9]. Previous work found that participants gave more correct responses in kanji construction tasks in the condition where they engaged in kūsho behavior (kūsho condition) compared to two other experimental conditions (i.e., rest and irrelevant movement conditions), as long as the visual feedback of their finger movements was available [9, 10]. In the present study, assuming the facilitative effect of kūsho behavior exists among older adults, we likewise used the kanji construction task and expected task performance in the kūsho condition to be better than in the other conditions.

In addition, the present study also analyzed the interaction of the kūsho effect and personal factors, including age, mini-mental state examination (MMSE) score, and years of education. It has been reported that cognitive task performance among older adults is highly influenced by factors such as educational level [16–19]. Aside from Itaguchi, Yamada (10), who found no reliable relation between vocabulary score and kūsho effect among young adults, no prior study has investigated the relation between demographic factors and kūsho effect. Using a naming task, one previous study did find that phonological cues facilitated task performance more so for older adults than for young adults [20], suggesting that an alternative network might facilitate recall for individuals with cognitive decline. We thus tentatively hypothesized that individuals among whom we assume cognitive decline is underway would show larger kūsho effects. Investigating such interactions is important for possible clinical and educational application of the behavior.

## Materials and methods

### Participants

Twenty-seven right-handed older adults (female = 21) participated in the experiment. One participant who could not precisely follow the experimental instructions was excluded from further analysis (see the Discussion for details). The average age of the analyzed participants ($n$ = 26) was 86.8 years old ($SD$ = 5.9). All were attending a senior citizen center in Tokyo, and the experiment was conducted in the center. The average score on the Edinburgh handedness inventory was 93.0 ($SD$ = 10.0). All participants preferred to use their right hand for writing and were native Japanese speakers who had completed elementary school in Japan without any apparent problems in reading and writing. We confirmed that all participants could see the visual stimuli on the display in the experimental environment.

Table 1 shows details about the participants' age, MMSE scores, and education. The amount education in years was self-reported. An MMSE, which is a 30-point questionnaire that is used in clinical and research settings to evaluate cognitive impairment, was conducted before or after the main experiment by one of the authors using a standard protocol. For two participants, the MMSE scores were below 24, which is the standard cut-off value for healthy and mild cognitive decline. However, note that we carefully assessed whether participants had sufficient cognitive capacity to understand and follow the instructions for the experiment by

**Table 1. Demographic data of participants.**

|  | Age | MMSE | Education (years) | Handedness |
|---|---|---|---|---|
| Average | 86.8 | 27.4 | 10.7 | 93.0 |
| SD | 5.9 | 2.5 | 2.7 | 10.0 |
| Max | 98 | 30 | 16 | 100 |
| Min | 71 | 21 | 6 | 70 |

checking the rationality of their verbal responses and the appropriateness of their behavior in the experiment and conversation.

For 12 of the participants, a vocabulary test known as 100 RAKAN [21], which comprises a 100-kanji-word reading, standardized based on visual and phonetic aspects of word familiarity, was also conducted to evaluate their vocabulary size. The minimum and maximum scores of the test are 0 and 100, respectively. Although we did not consider the vocabulary data in the analysis because several participants did not perform the test, the scores for these 12 participants are available in the Supporting Information.

## Kanji construction task

In the kanji construction task, participants were asked to verbally identify a kanji character based on three kanji subparts presented on a computer display [6, 9, 10], which was about 50 cm away from the participants. Before stimulus presentation, a fixation cross appeared for 1 s on the center of the display. After the fixation cross disappeared, three kanji subparts, each about 6 cm × 6 cm, were presented on the center of the screen in a triangle formation (Fig 1a). The kanji parts remained on the display for 10 s, and answers were not accepted after that time. Participants were instructed to say the original character aloud into a microphone placed in front of them as soon as they came up with the answer. The verbal responses were also recorded using a portable audio recording device during the experiment to check the responses and calculate response time later.

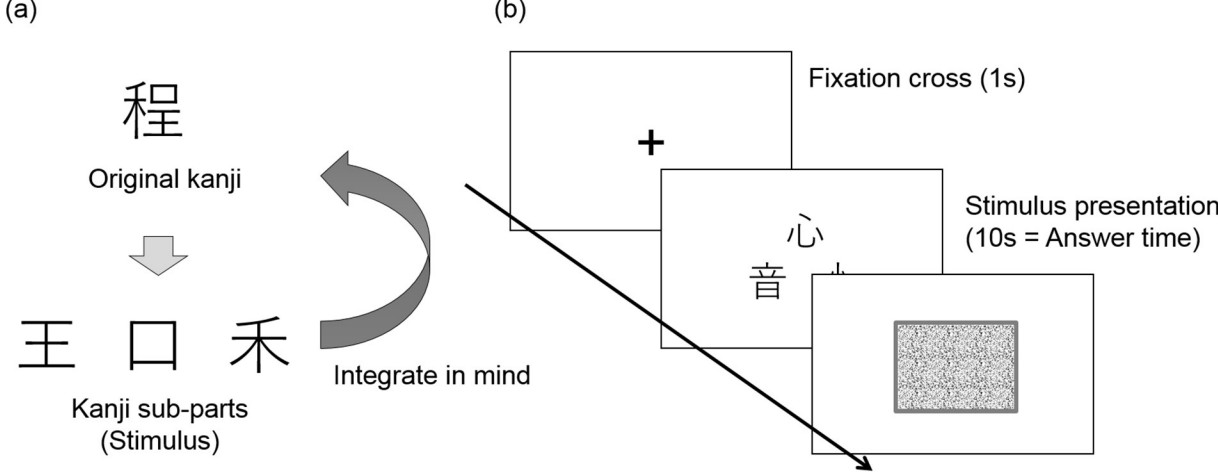

**Fig 1. Experimental stimuli and sequence.** (a) An example of an original kanji character and its subparts. In the task, participants tried to integrate one original character from three decomposed subparts presented in a triangle arrangement. In the above example, the right subpart has to be horizontally compressed before arranging the three subparts and constructing the original kanji character. (b) The time course of one trial in the kanji construction task.

To form the kanji characters in their minds, participants were allowed to overlap, expand, or reduce the presented subparts. In Fig 1a, for example, to make the original kanji character, one subpart was placed on the left side and horizontally compressed, and the other subparts were vertically compressed and arranged on the right side. This type of puzzle is popular in Japan (for example, they sometimes appear in TV programs, albeit without any instructions on hand or finger movements), and all participants understood the rules without difficulty. We judged a response correct when the participant's response matched the reading of the original kanji. A response was judged incorrect when it was wrong or not given within the specified answering period. Response times were not statistically tested or discussed because participants' voices were not always loud or clear enough to detect the onset of the responses, and the ratio of missing values was very high (31% of correct responses were missing values for response time, and 69% of the data were used to calculate response times).

We used three experimental conditions based on previous studies [9, 10] to test the facilitative effect of kūsho behavior on kanji construction task performance. In the kūsho condition, participants had to move their right index finger freely in a writing manner on a table. It is known that the kūsho effect is greater when the index finger touches a table surface than when it writes in the air [6]. Here, participants were encouraged to use kūsho behavior (finger writing) to solve the task. In the static condition, participants kept their right hand clenched on the table and were not allowed to move any fingers. In the circle-drawing condition, participants made circular motions with their right index finger on the table surface during the trial. In all conditions, the left hand always stayed clenched on the table. The static condition was taken as the control condition, which involved neither motor planning nor visually or kinematically meaningful finger movement feedback. The circle-drawing condition was introduced to eliminate the possibility that simply moving a finger while thinking would benefit task performance, under the assumption that while it involved motor planning and motion-related feedback, it was irrelevant to writing kanji characters.

Participants were instructed to turn their face away from the screen and watch their right index finger or hand in a trial to elicit the facilitative effect of kūsho behavior, since people do not always watch their finger when they are doing kūsho. If they turned their eyes away from the finger, one of the experimenters gave an instruction to watch the finger. Our previous research, using two visual conditions (eye-on-hand and eye-on-display conditions), found that the visual feedback of kūsho finger movement was key to triggering the facilitative effect of the behavior [9, 10]; that is, no benefit is derived if one does not see the finger movement (eye-on-display condition). Two experimenters watched the participants' finger movements as well as the focus of their gaze in the experiment to confirm that they followed the instructions, one from the side and the other one from the front of the participant. Only the eye-on-hand visual condition was used because our main purpose was to demonstrate the facilitative effects of kūsho behavior and examine the interaction between the kūsho effect and personal factors.

The participants carried out 60 trials in total (20 trials × 3 conditions). The experimental condition was performed as a block, and the order of conditions was counterbalanced among participants. Before starting the main trials, participants performed several practice trials without instructions for finger and hand conditions to observe whether they spontaneously showed kūsho behavior. We then explained the experimental conditions, and the participants again performed practice several trials to ensure that they followed the instructions.

## Stimulus

We used the same sets of kanji stimuli as in previous studies [9, 10], which included a total of 60 kanji characters. All selected kanji characters were broken down into three subparts (Fig 1a).

Only one actual kanji character could be constructed from the three subparts (i.e., there was one correct answer for each trial). The sub-parts were placed in a unique arrangement, different from the original one so that it would not be too easy to find the correct answer. The subparts in each stimulus were always presented in the same relative position. For the three experimental conditions, three sets of kanji characters were created to have the same character properties on average in terms of familiarity, complexity, grade level learned in school, and difficulty (for further details of the stimuli, see [10]). The three sets were randomly assigned to each hand condition. All characters used in the present study are learned in primary school in Japan. We confirmed that the participants knew and were able to read the characters after the main experiment.

## Analysis

To examine the effects of the hand condition, a one-way ANOVA was performed on the number of correct responses, and as a post hoc analysis, we conducted multiple comparisons using Shaffer's method. We also calculated the partial correlation coefficients between two variables of the kūsho effect, age, MMSE score, and education. Kūsho effect was defined as the difference in the number of correct responses between the kūsho and static conditions. Partial correlation was used because it calculates the correlation coefficient between two target variables with the effects of other variables removed. To repeat the correlation analyses, we applied Bonferroni correction to control type I error probability. We analyzed four factors in the correlation analysis; thus, the total number of tests was six.

## Ethics statement

This study was conducted in accordance with the principles in the Declaration of Helsinki. The study and consent procedure were approved by the Ethics Committee on Human Research of Waseda University. All participants provided their written informed consent.

## Results

### Facilitative effect of kūsho behavior

We should first note that all participants (26 out of 26) spontaneously showed writing movements with their index finger while thinking during the practice trials without instruction. Although these movements were not video recorded, the two experimenters agreed that kūsho behavior was evident in all observations.

The experimental results for the kanji construction tasks confirmed the beneficial effect of kūsho behavior for older adults. The average numbers of correct responses were 5.92 ($SD$ = 3.22) in the kūsho condition, 4.42 ($SD$ = 3.36) in the static condition, and 4.65 ($SD$ = 3.32) in the circle-drawing condition (Fig 2). ANOVA showed the significant effect of the experimental condition ($F_{(2, 50)}$ = 7.22, $p$ < .01, $\eta_p$ = 0.22). Multiple comparisons using Shaffer's correction revealed that the number of correct responses in the kūsho condition was significantly larger than in the static condition ($t_{(25)}$ = 4.02, $p$ < .05, $d$ = 0.80) and the circle-drawing condition ($t_{(25)}$ = 2.61, $p$ < .05, $d$ = 0.52). There was no statistical difference in the number of correct responses between the static and circle-drawing conditions ($t_{(25)}$ = 0.23, $p$ = .58, $d$ = 0.11). The average response times were 6.9 s ($SD$ = 1.2) in the kūsho condition, 6.5 s ($SD$ = 1.8) in the static condition, and 6.8 s ($SD$ = 1.3) in the circle-drawing condition.

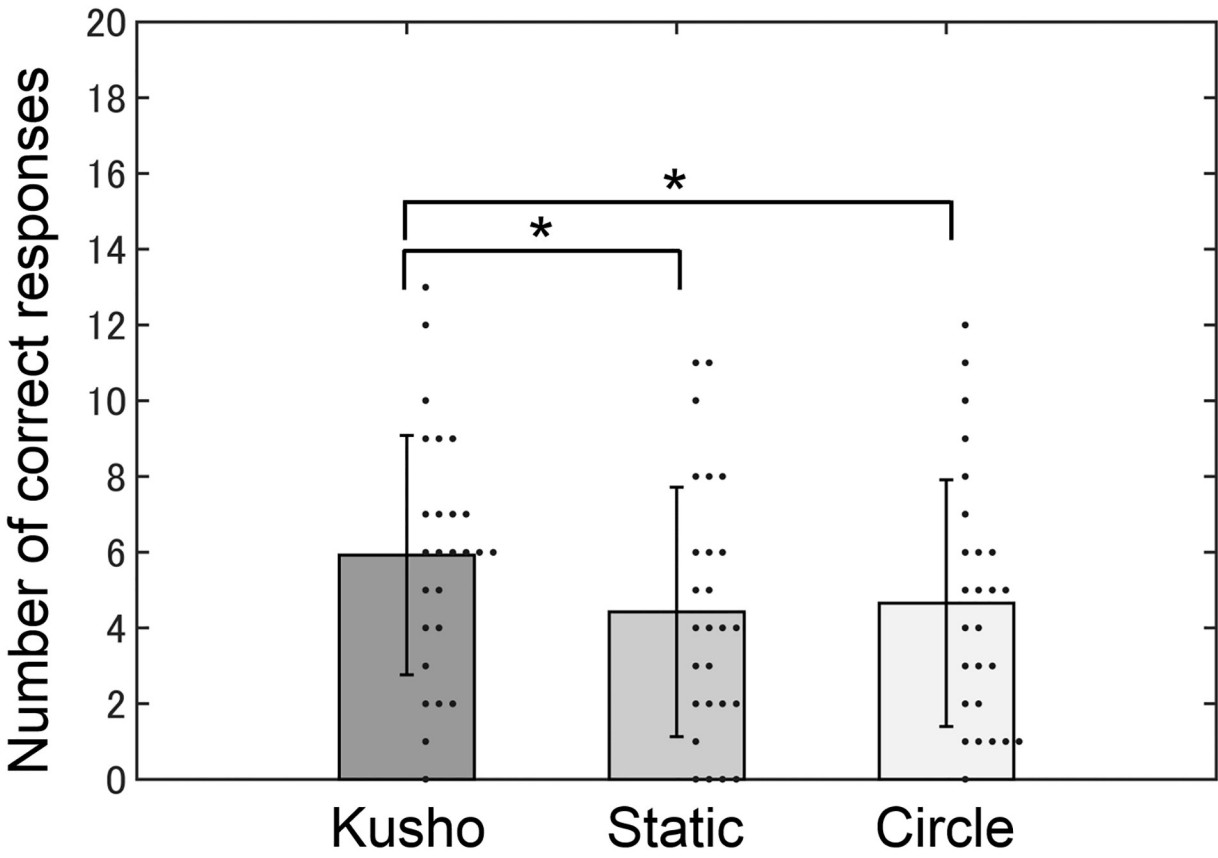

**Fig 2. The number of correct responses in the kanji construction task.** A dot indicates the number of correct responses for a single participant in each condition. An asterisk indicates that there is a statistically significant difference in multiple comparisons with an alpha level of 5%.

## Correlation analyses of the kūsho effect

Partial correlation analysis revealed that the kūsho effect was negatively correlated with years of education but not with the other two factors. Table 2 shows the partial correlation matrix. The correlation coefficients of the kūsho effect were -.11 for age ($t(23) = 0.51$, $p = .99$), -.06 for the MMSE ($t(23) = 0.28$, $p = .99$), and -0.63 for education ($t(23) = 3.75$, $p < .05$) with Bonferroni correction. Even when excluding one outlier whose age was 71, the correlation coefficients for the kūsho effect and education were still significant ($r = .60$, $p < .05$), though the correlation coefficients for the kūsho effect and age turned out to be larger ($r = .38$, $p = .47$).

**Table 2. Correlation matrix for the number of correct responses in the three experimental conditions and personal factors.**

|  | Experimental conditions | | | Personal factors | | |
|---|---|---|---|---|---|---|
|  | **Kūsho** | **Static** | **Circle** | **Age** | **MMSE** | **Education** |
| Kūsho | 1 | .83 | .67 | .30 | .44 | .15 |
| Static |  | 1 | .80 | .37 | .52 | .51 |
| Circle |  |  | 1 | .39 | .21 | .34 |
| Age |  |  |  | 1 | .03 | .28 |
| MMSE |  |  |  |  | 1 | .15 |
| Education |  |  |  |  |  | 1 |

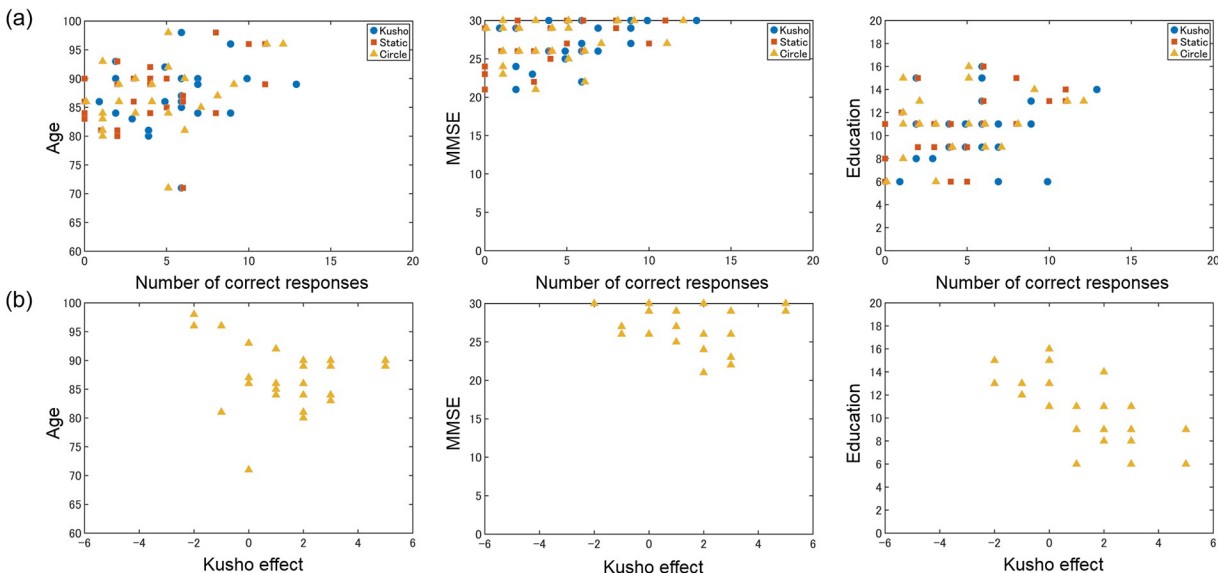

**Fig 3.** (a) Scatter plots of the number of correct responses in the three experimental conditions against the three personal factors. (b) Scatter plots of the kūsho effect against the three personal factors.

Fig 3 shows scatter plots for the three personal factors and for the kūsho effect. There is a consistent trend between the kūsho effect and education (right panel of Fig 3B) but not in the other pairs. We recognize a ceiling effect in the MMSE score (middle panel of Fig 3B) and one outlier in the scatter plot for age (left panel of Fig 3B).

As supplementary information for future reference, Table 3 shows a descriptive correlation matrix for the number of correct responses and other personal factors without statistical testing. For the current participants, we obtained mainly three results from the matrix. First, task performance among the three experimental conditions was well correlated ($r = .83$, $r = .67$, and $r = .80$). Second, the magnitudes of the correlation coefficients among the personal factors were relatively low in general. Third, both MMSE and education were moderately correlated with the baseline task performance (the coefficients for the static condition were $r = .52$ for MMSE and $r = .51$ for education).

## Discussion

The present study mainly obtained two novel findings. First, we confirmed that kūsho behavior improved cognitive task performance that required visual processing of written language

**Table 3. Partial correlation matrix of the kūsho effect and the other three factors (n = 25).**

|  | Age | MMSE | Education years |
|---|---|---|---|
| Kūsho effect | -.11 (.00) | -.06 (-.08) | -.63* (-.60*) |
| Age |  | -.01 (.11) | .04 (.38) |
| MMSE |  |  | .18 (.07) |

* Asterisk indicates significance of the correlation coefficient with Bonferroni correction ($p < .05$). MMSE: Mini-mental state examination.

among older adults. This result is consistent with previous studies that found facilitative effects of kūsho behavior among university students [6, 9, 10]. We also observed that kūsho behavior occurred spontaneously in all participants without being given instructions. Second, by analyzing the partial correlations between the kūsho effect and demographic factors, we found a significant linear inverse relationship only between the kūsho effect and years of education. While we did not expect that only education would be related to kūsho effect, the overall results were generally consistent with the previously proposed idea that kūsho behavior has a beneficial influence on cognitive processing for written language. Such benefits could derive from lessening a cognitive burden that individuals with more education could manage without finger movement.

## Facilitative kūsho effects in older adults

The present study confirmed the benefits of kūsho behavior in older adults, thus supporting our main hypothesis, while previous studies showed its benefits in young adults [9, 10]. We should note, however, that the total number of correct responses differed between previous studies and the present one (9.7 vs. 4.4, respectively, in the static condition on average). Moreover, the effect sizes for kūsho behavior were somewhat larger in the present study—Cohen's *d* was 0.48 in the previous study [10] and 0.80 in the present study. Since task performance in the circle-drawing condition was lower than in the kūsho condition, the kūsho effect was not attributed to mere finger movement but specifically to writing-based finger movement. Although the present experiment did not address the effect of the *visual feedback* of finger movement, we observed improvements in cognitive task scores due to kūsho behavior, concurring with the idea that the visual feedback of finger movement helps cognitive processing [6, 9, 10]. Taken together with previous studies, the present results suggest that the interaction between language and sensorimotor systems for the facilitative effect of kūsho behavior is still working, at least among kanji-culture older adults.

The correlation analyses suggest that the benefits of kūsho behavior for cognitive tasks do not decline with age. The partial correlation coefficient between kūsho effect and age was -.11 at a statistically insignificant level, suggesting there might not be a consistent relation between them. Although the scatter plot in Fig 3B appears to show a negative correlation between age and kūsho effect, the shape of the plot is likely attributable to other confounding factors (in this case, years of education). In the current study, we did not control age-related variables other than MMSE scores and years of education, which reflect general cognitive ability and some intellectual factors, respectively. The present results thus imply that age-related decline in perceptual and motor performance (e.g., low-level vision processing, kinesthetic and touch sensation, and finger dexterity) does not play a critical role in the facilitative effect of kūsho. It should be noted that most participants were older than 80 (average age: 86.8), and we cannot exclude the possibility that the age range of the participants contributed to the low correlation between age and kūsho effect.

One might argue that the kūsho effect is not facilitative but inhibitory because of the tight coupling between the visual and sensorimotor representation of the characters. It is possible to assume that the "baseline" is not the static condition but the kūsho condition, and consequently the constraint on finger movement in the static condition might have an inhibitory effect, especially for individuals with lower cognitive function due to less education. While this hypothesis does not have direct evidence, it is not absurd. Previous studies and our own data have shown that, among Japanese people, kūsho behavior occurs frequently in both daily life and experimental settings without any instructions, regardless of age and time period [6, 9]. Such spontaneous kūsho behavior should be distinguished from the strategy used by pure

alexia patients where they read letters or characters by tracing them, sometimes called "kinesthetic facilitation" [22–25]. However, both types of finger writing appear to stem from a tight coupling of visual representation and kinetic factors in written languages, as supported by studies in neuropsychology [26, 27] and brain imaging [28]. In line with such coupling, one participant who was excluded from the analysis in the present study (age = 81, MMSE = 22, years of education = 16) showed a strong tendency to use kūsho behavior even in the static condition—that is, even when he was not supposed to use kūsho, he could not stop moving his index finger. These observations might indicate tight coupling between finger actions and written characters [14, 29], suggesting that a large load would be required to inhibit finger movement among individuals who have acquired it as a natural habit.

## Education years, MMSE score, and other possible variables for the kūsho effect

Different sizes of the kūsho effect have been found in various studies; this might be explained by the fact that participants of the studies had different years of education. In this study, we found a moderate-sized negative correlation between years of education and the kūsho effect; individuals with less education showed larger kūsho effects than those with more education. This is consistent with our previous research [10]. That is, if we simply assume education-dependent changes in kūsho effect, the different sizes of the kūsho effect between the studies (0.48 vs 0.80) can be explained without assuming age-dependent changes in the kūsho effect. The young adults in the previous study were university students, and therefore their education level was higher than that of the participants in the present study (at least 12 years, vs. 10.7 years on average). Accordingly, the kūsho effects for a participant group that includes various educational levels should be higher than those of university students.

Specific cognitive declines are likely to intervene in a larger kūsho effect, although this study did not clarify the exact factors and mechanisms that are modulated by years of education. Education level is known to positively correlate with various cognitive [17, 18, 30, 31] and brain functions [16, 32] in older adults and has also been used as a marker of the concept of cognitive reserve, which refers to susceptibility to age-related changes in cognitive functions [19, 33–35]. These previous findings together suggest that the individuals with more years of education in this study might have produced a moderate number of correct responses even in the condition where the kūsho behavior was not available, and, therefore, they benefited less from kūsho behavior than those with fewer years of education. In the present study, the MMSE scores, which reflect general cognitive decline, was not related with education or the kūsho effect, suggesting that the increase in kūsho effect was not attributed to global cognitive status, but to a decline in rather specific cognitive functions. One possible candidate to explain this could be the high level of visual processing. Previous studies have suggested that kanji construction tasks strongly activate visual areas [36] and that visual feedback from kūsho movement helps with solving tasks [9, 10]. This requires further investigation using comprehensive cognitive tests such as the WAIS-III. Another candidate may be vocabulary size; vocabulary scores are known to be positively correlated with education years [37–39]. Although vocabulary score has been suggested as not being associated with the kūsho effect in younger adults [10], there is still a possibility that it interacts with the sensorimotor processing in older populations. Kinesthetic feedback of kūsho finger movement, which was shown to have only little (non-significant) effect on task performance if visual feedback of movement is not available in younger adults [9, 10], could also contribute to the facilitation effect in older adults, as the importance of visual feedback in task processing was not directly tested in the present study. Collectively, previous findings and our data suggest

that decline in a specific function may be involved with a modulation of the kūsho effect in the kanji construction task.

While this study is the first to demonstrate the beneficial effects of kūsho behavior for older adults, some limitations should be noted. First, the observed effect of kūsho behavior is quite task-specific [9]. It is thus not clarified whether kūsho effects occur with other kinds of tasks. Second, although it has been shown that visual feedback of finger movement is critical to inducing the benefits of kūsho behavior, this study used only one visual condition instead of comparing two visual conditions. Therefore, we do not have direct evidence for the importance of visual feedback for older adults. Third, the sample size and age range were relatively small in terms of generalizing the findings to the whole population. That said, the sample size was adequate for detecting a middle to large effect size (*power* = .70 for $f^2$ = 0.15 and *power* = .97 for $f^2$ = 0.35) in the ANOVA, and the correlation was reliable (*r* = -.63). Twenty-six older adults participated in the present experiment, and their average age was 86.8 years. Given the huge variability in the characteristics of older adults, a larger sample, one that includes younger older adults, may be needed to more firmly establish our findings.

Even with these limitations, the present results can provide some general implications. First, kūsho behavior could have clinical and educational benefits for individuals with a decline or a lack of ability in specific cognitive functions such as stroke patients and second-language learners [10]. As mentioned earlier, alexia patients can sometimes recognize words and characters by tracing them with their index finger [22–25, 27], which is essentially the same behavior as kūsho. Thomas (12) also reported that kūsho behavior improved kanji learning for second-language learners of Japanese. Our finding that the kūsho effect was negatively related to years of education also suggests that kūsho is a helpful tool, especially for individuals with a declining specific, but not general, cognitive function. Accordingly, although kūsho behavior could be beneficial for individuals with mild cognitive impairments, it may not correlate with MMSE score. Second, the kūsho effect observed in this study may provide an instantiation of embodied cognition. It has been shown that finger movement differently influences arithmetic problem solving, depending on how calculation strategies are acquired [40], and the interference of finger movement is culturally modulated [41]. Our data suggest that demographic factors such as intellectual ability and age should be included when considering the interaction between bodily movement and cognition.

## Supporting information

**S1 Table. Raw data obtained in the experiment.**
(XLSX)

## Author Contributions

**Data curation:** Yoshihiro Itaguchi, Chiharu Yamada.

**Formal analysis:** Chiharu Yamada.

**Funding acquisition:** Yoshihiro Itaguchi, Kazuyoshi Fukuzawa.

**Supervision:** Yoshihiro Itaguchi, Kazuyoshi Fukuzawa.

**Writing – original draft:** Yoshihiro Itaguchi.

**Writing – review & editing:** Yoshihiro Itaguchi, Chiharu Yamada, Kazuyoshi Fukuzawa.

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
