## [Decision Letter · Decision Letter 0]

25 Jul 2019

PONE-D-19-16768

Writing in the air: facilitative effects of finger writing in older adults

PLOS ONE

Dear Dr. Itaguchi,

Thank you for submitting your manuscript to PLOS ONE. After careful consideration, we feel that it has merit but does not fully meet PLOS ONE’s publication criteria as it currently stands. Therefore, we invite you to submit a revised version of the manuscript that systematically addresses all the points raised by the three reviewers during the review process (see below).

We would appreciate receiving your revised manuscript by Sep 08 2019 11:59PM. To enhance the reproducibility of your results, we recommend that if applicable you deposit your laboratory protocols in protocols.io, where a protocol can be assigned its own identifier (DOI) such that it can be cited independently in the future. For instructions see: http://journals.plos.org/plosone/s/submission-guidelines#loc-laboratory-protocols

We look forward to receiving your revised manuscript.

Kind regards,

Emmanuel Manalo, PhD

Academic Editor

PLOS ONE

**Journal Requirements:**

2. Thank you for stating that “The funders had no role in study design, data collection and analysis, decision to publish, or preparation of the manuscript” in your financial disclosure.

Please also provide the name of the funders of this study (as well as grant numbers if available) in your financial disclosure statement.

**Additional Editor Comments (if provided):**

All three reviewers acknowledge the possible useful contribution that your research could make to this topic area, but they also list numerous issues/errors that need to be addressed before this paper could be considered as suitable for publication. Note in particular the need to be more accurate in word use and claims about what your research findings show, and numerous issues pertaining to the method. Please carefully and systematically address all of these issues in revising your paper. In your response letter, please make clear how you have addressed each and every one of them.

**Comments to the Author**

1. Is the manuscript technically sound, and do the data support the conclusions?

Reviewer #1: Partly

Reviewer #2: Yes

Reviewer #3: Yes

2. Has the statistical analysis been performed appropriately and rigorously? 

Reviewer #1: I Don't Know

Reviewer #2: Yes

Reviewer #3: Yes

3. Have the authors made all data underlying the findings in their manuscript fully available?

Reviewer #1: No

Reviewer #2: Yes

Reviewer #3: Yes

4. Is the manuscript presented in an intelligible fashion and written in standard English?

Reviewer #1: Yes

Reviewer #2: Yes

Reviewer #3: Yes

5. Review Comments to the Author

Reviewer #1: This small study reports intriguing findings, and contributes to the stream of on-going work on kūsho. The general design of the study is sound. However, to my mind it somewhat overstates its findings, and in several places leaves out critical information. Specifically:

(1) I would suggest that the abstract be revised to include greater specification of the meaning of the terms ‘kūsho’ and ‘the kūsho effect’. I showed this abstract to several colleagues who have no exposure to kanji cultures. They were wholly mystified as to the meaning of ‘writing in the air or on a surface with one’s index finger’. Unless you anticipate addressing only a readership that is already familiar with kūsho, I think you need to add to the Abstract some further identification of its meaning.

(2) Line 66: ‘Individuals with demographic factors related to cognitive decline’ is both vague and disguises an inference you are making. Can you be more straightforward? How about ‘older adults, among whom we assume cognitive decline is underway’?

(3) Line 67: This raises a major problem: from here on out you refer repeatedly to ‘larger [/ smaller] kūsho effects’. You don’t actually measure the SIZE of the effect, but rather only the frequency of its presence / absence. This comes to a head in the interpretation of Figure 3b, where the horizontal axis records the kūsho effect from –6 to (+)6. Presumably, this is not a measure of its ‘larger / smaller’ SIZE, but rather its frequency of use—correct? This needs to be clearly specified. (I would add that Figure 3 is out of focus and difficult to read. Can a new version of it be created, with higher resolution?)

(4) Line 82. Please provide full details on the content of the MMSE.

(5) Lines 83–5. Please also explain how you ‘carefully assessed whether participants had sufficient cognitive capacity…’

(6) The authors declare under ‘Data availability’ that ‘All relevant data are within the manuscript and its Supporting Information files.’ However, I would like to see the full set of 60 experimental stimuli, and do not find them in the materials made available to me in the course of this review.

(7) Line 16 mentions that ‘This type of puzzle is popular in Japan’. Could you elaborate? If kanji-assembly tasks constitute a familiar puzzle or pastime, how is that carried out? Does kūsho play any role in it—a role that might carry over into your results?

(8) Lines 130–1. How, exactly, did you ‘encourage [participants] to turn their eyes from the screen to their right index finger’? Did any participants resist doing so, fail to do so, or fail to maintain their gaze on the kūsho-producing hand?

(9) Discussion section. I think this section would benefit for a more conservative statement of your findings. For example, have you really ‘confirmed that kūsho behavior improved cognitive task performance’ as opposed to ‘confirmed that kūsho behavior co-occurs with improved cognitive task performance, relative to performance under the other two test conditions’? Likewise, have you shown that ‘kūsho behavior has a beneficial influence on cognitive processing’ as opposed to that ‘kūsho behavior correlates with greater success at integrating parts of kanji into familiar whole orthographic units’?

Similarly, in lines 330–2, the authors claim that because of the negative relation between kūsho and years of education, ‘our findings…suggest that kūsho is a helpful tool, especially for individuals with declining cognitive function’. I’m not sure you can draw that inference, rather than simply that ‘kūsho is less frequently employed by older individuals with more years of education relative to their age-mates with less education’. I don’t think you’ve shown that less education correlates with greater cognitive decline.

(10) Lines 251–8. You make the case for visual feedback as a prerequisite for the efficacy of kūsho behavior. But how do you deal with the commonly-observed spontaneous use of kūsho in settings where visual feedback is seemingly deliberately unavailable, such as writing on the top of one’s thigh underneath a desk top, or in the air with the writer’s gaze turned away from the kūsho-producing hand? And if visual feedback were necessary to induce the kūsho effect, why would you have to ‘ENCOURAGE [participants] to turn their eyes from the screen to their right index finger’ as opposed to simply relying on their spontaneous integration of visual input to the kūsho effect?

(12) As conceded in lines 318–9, the sample size is quite small. I would expect that with a larger pool of participants, the effects of age and years of education might be more fully displayed. I would recommend that the study be augmented.

There are a few typos, or sites where word choice might be improved.

Lines 22, 25: Missing definite article before ‘__ kūsho effect’

Line 22: Since the participants’ education is not assessed qualitatively or with respect to its content, I would suggest ‘…(MMSE) score, and number of years of education’

Reviewer #2: This study demonstrates the beneficial properties of “Kūsho behavior” among Japanese older adults in identifying kanji characters from disjointed units of the characters. The study was based on an earlier investigation employing the same experimental design with younger Japanese persons. The authors report two main findings of their investigation: First, the existence of a facilitatory behavior in older persons to achieve correct identification of kanji characters by deployment of Kūsho behavior and second, the negative association of this phenomenon with the educational level of the participants.

Although, these findings are mainly relevant for kanji writing-system users, it is an interesting investigation that may add new knowledge to the aging literature. However, there are some concerns that need to be addressed before the paper can be published.

1) Abstract: it needs to be rewritten after correcting the manuscript. Specifically the last sentence should be corrected. It cannot be asserted that “The study confirms the benefits of finger movement for “solving cognitive tasks….”. This study deals with the effect of “Kūsho behavior” on identification of written language/kanji characters and not with various cognitive tasks. In addition, it should be stated that: “the kūsho effect may be mediated by “education” and not by “intellectual ability”.

2) Introduction:

a. My main concern is related to the rationale of assessing the facilitative effect of “Kūsho” in older adults. The authors argue that they wish to test whether the “Kūsho” behavior is universal, regardless of age. However, if this phenomenon occurs due to the written language system, i.e., use of kanji, the phenomenon should be present on every literate person, disregarding the age of the individual. Thus, the authors do not provide a reason for why this phenomenon should not exist in older literate people. I believe that the interest to study older adults in this experimental situation is to evaluate whether “Kūsho behavior” is equally relevant for this population as it is for younger individuals. My understanding is that any literate person in the kanji-system may rely at different degrees on “Kūsho behavior”. If this is true, the reason to assess older persons is to better understand how normal age-related changes in sensorimotor systems and cognitive functions affect “Kūsho behavior”.

b. It would be a strength if the authors add information on “cognitive reserve” (CR), especially in view of their findings on education. CR is particularly important to understand age-related changes, both declines and gains, in late adulthood in any culture and to my knowledge across different languages.

c. It would be important to develop more on why vocabulary was not addressed in this investigation. It is well known that vocabulary can be highly important in determining outcomes of language studies and aging. It would be more advisable to actually add vocabulary data to this manuscript. See more comments on this issue below.

3) Methods:

a. It seems that the MMSE cut-off of 24 was not use as inclusion or exclusion criteria, which is understandable due to the advanced age of many of the participants. However, the authors claim to have examined a sample of healthy older adults. This cannot be stated since there are participants scoring below MMSE 24 who might have important cognitive declines possibly related to dementia or mild cognitive decline. This issue needs to be addressed and acknowledged.

b. It would strength the study to present data on vocabulary in case that such data are available. It would be important to show whether education, vocabulary, or both exert an effect on less deployment of Kūsho behavior in older adults.

c. Why is handedness measured and reported? The authors do not mentioned this variable in either the introduction or the discussion. If it is no of importance, it should be removed.

4) Results:

a. Related to the previous comment, it would be interesting to compare the relative importance of education vs. vocabulary. Findings would enhance the interest of the study.

5) Discussion:

a. If vocabulary data were available, it would be necessary to expand the discussion on p. 11 to cover “Educational level vs vocabulary level and Kūsho effect”.

b. This part should also be amended by including a discussion on CR and the final findings that in the best of cases comprise both education and vocabulary.

Reviewer #3: Authors have conducted a series of experiments investigating kusho behaviors. Now, they extend their target to elderly people living in Japan.

I appreciate their new contribution using elderly people. However, their research question is not clearly written in the current manuscript. For that reason, I suppose, their discussion also has not been deepened sufficiently enough.

I would point out two main negatives of this paper:

(i) No comparisons with younger participants

(ii) No further examinations of cognitive abilities specifically, memory.

I suspect that these negatives might come from the fact that they had no valid hypothesis when they started this study.

To compensate these negatives, authors should work even harder. I will try to provide them inspirations for that as follows.

Major points

(1) Motivation of the study.

I have to say that authors did not successfully explain their motivation to study elderly people. I know they stated:

While kūsho behavior is widely observed among Japanese people regardless of age, no previous study has specifically confirmed the effect of kūsho behavior on cognitive processing among older adults. The present study, therefore, aimed to investigate whether the facilitative effect of kūsho behavior is universal, regardless of age, at least among kanji-culture adults. (lines 41-44)

This does not sound intrinsic. I would like authors to more clearly ensure accountability for the employment of elderly people for this study.

I will give some suggestions regarding this point (although I do not mean to force authors to discuss these; they would achieve better than me.)

(i) What is the main influence of the aging on kusho behaviors? No effects? Reducing or amplifying?

(ii) The computer environment of elderly people when they learned kanji was greatly different from the modern one. In a sense, they evaded the use of electric devices during their childhood. Then, the kusho effects might be genuinely observed among them as compared to younger participants.

(iii) In spite of the memory decline generally seen among the elderly, the kusho movements would be stably maintained. Even more, it may facilitate the memory of kanji.

(2) Dementia and MCI

It was good that authors examined MMSE. I found by Table 1 and Figure 3 that they included participants who were categorized as MCI or even as dementia (score 21). Authors should discuss the effects of MCI/dementia on kusho.

(3) ANCOVA

I think they could use ANCOVA instead of ANOVA for their analysis, including age, MMSE score and years of education as the covariates.

(4) Paragraph writing

I would suggest that authors rewrite all paragraphs to ensure that a conclusion statement comes to the first line and then the explanations follow in the rest of the paragraph.

Minor points

(5) Lines 36-37, “kanji writing-system users”: this expression is inappropriate because Chinese speaking people do not call their character kanji. Instead, authors may just want to use “character users” or “people who use a writing system with characters”.

(6) Participants: did all participants prefer to use their right hand for writing? Authors should clearly state this point.

(7) Experimental settings: how far was the computer from a participant? How many experimenters stayed beside a participant and what locations? Authors should explain how they observed participants’ responses sufficiently.

(8) Stimuli: how did they decide the locations of subparts in the triangle? Did authors avoid a subpart to write first in the original kanji came to the top of the triangle or never mind and put it randomly?

6. PLOS authors have the option to publish the peer review history of their article (what does this mean?). If published, this will include your full peer review and any attached files.

Reviewer #1: Yes: Margaret Thomas

Reviewer #2: No

Reviewer #3: No

---

## [Author Response · Author response to Decision Letter 0]

21 Aug 2019

The comments from the reviewers have been responded in an attached file.

---

## [Decision Letter · Decision Letter 1]

23 Sep 2019

PONE-D-19-16768R1

Writing in the air: facilitative effects of finger writing in older adults

PLOS ONE

Dear Dr. Itaguchi,

Thank you for submitting your manuscript to PLOS ONE. After careful consideration, we feel that it has merit but does not fully meet PLOS ONE’s publication criteria as it currently stands. Therefore, we invite you to submit a revised version of the manuscript that addresses the points raised during the review process.

All three reviewers acknowledge that you have improved the manuscript this time round. However, all three reviewers also provide point-by-point details of issues that remain unsatisfactorily resolved. Therefore, I would like you to very carefully and systematically address all those issues/points they have raised in a further revision of this manuscript.

We would appreciate receiving your revised manuscript by Nov 07 2019 11:59PM. To enhance the reproducibility of your results, we recommend that if applicable you deposit your laboratory protocols in protocols.io, where a protocol can be assigned its own identifier (DOI) such that it can be cited independently in the future. For instructions see: http://journals.plos.org/plosone/s/submission-guidelines#loc-laboratory-protocols

We look forward to receiving your revised manuscript.

Kind regards,

Emmanuel Manalo, PhD

Academic Editor

PLOS ONE

Reviewers' comments:

Reviewer's Responses to Questions

**Comments to the Author**

1. If the authors have adequately addressed your comments raised in a previous round of review and you feel that this manuscript is now acceptable for publication, you may indicate that here to bypass the “Comments to the Author” section, enter your conflict of interest statement in the “Confidential to Editor” section, and submit your "Accept" recommendation.

Reviewer #1: (No Response)

Reviewer #2: All comments have been addressed

Reviewer #3: (No Response)

2. Is the manuscript technically sound, and do the data support the conclusions?

Reviewer #1: Partly

Reviewer #2: Yes

Reviewer #3: Yes

3. Has the statistical analysis been performed appropriately and rigorously? 

Reviewer #1: I Don't Know

Reviewer #2: Yes

Reviewer #3: Yes

4. Have the authors made all data underlying the findings in their manuscript fully available?

Reviewer #1: Yes

Reviewer #2: Yes

Reviewer #3: No

5. Is the manuscript presented in an intelligible fashion and written in standard English?

Reviewer #1: Yes

Reviewer #2: Yes

Reviewer #3: No

6. Review Comments to the Author

Reviewer #1: The revisions of this ms. raise its quality, but feel a bit superficial. My objections number (10) is smoothed over without really resolving it; (11) is sidestepped; and most of the rest appropriately dealt with. However, I don't think the authors have really dealt with a matter raised by Reviewers #2 and #3 , "No comparison with a younger cohort". Do Reviewers #2 & #3 feel that this has been adequately addressed? If so, I would reluctantly shift my assessment to "Accept".

Reviewer #2: The authors reasonably addressed my previous concerns and they have now delivered a much improved manuscript. However, there are still some minor points that need to be corrected:

1. In the abstract, I suggest to better clarify the purpose of the investigation. Instead of: “This study investigated whether kusho could help older adults solve a cognitive task” it should be:

“This study investigates whether the facilitative effect of kusho is affected by old age”.

2. Supporting information: I understand that due to the missing data on vocabulary the authors are not able to address the role played by this variable and therefore they provided few existing results as supplementary material. Though, they have to explain the sense of these data. In other words, whether higher scores reflect higher lexical skills.

3. Discussion p. 12:

a. The first sentence on the 2nd paragraph needs to be rephrased (lines 324 to 326). The meaning is not clear to me.

b. Lines 334 and 335: It should be “global cognitive status” and not “general cognitive decline”.

c. Line 335: The authors claim that a possible candidate mediating the effects of education is visual

processing. However, it is largely known that older adults suffer normal declines not only on their visual acuity, but also on their perceptual visual abilities (see e.g., Monge & Madden, 2016). Importantly, no visual evaluation was conducted in this study. Thus, I would rather suggest thinking about the advantages of well-learned proprioceptive functions in this population. Some authors have suggested that no age effect exists in proprioception when older adults know in advance the movements related to a specific action (Stelmach & Sirica, 1986).

Monge, Z. A. & Madden, D. J. (2016). Linking cognitive and visual perceptual decline in healthy aging: The information degradation hypothesis. Neuroscience and Biobehavioral Reviews, 69, 166-173.

Stelmach, G. E. & Sirica, A. (1986). Aging and Proprioception. Age, 9, 99-103.

Reviewer #3: Major points.

I recognized that the authors improved the manuscript greatly. However, I still have an uncomfortable feeling against their logics. I will indicate two points; their aim of the study and their way of description.

Their aim of the study:

First, I would categorize two types of papers.

(1) Papers that reveal a new phenomenon.

(2) Papers that investigate a mechanism of a phenomenon.

This paper seems to belong to the first one, with having a factor of the second. It revealed that the elderly people in Japan showed kusho behaviors in a similar way as having shown in younger generations, and as well investigated the relationship with individual parameters including years of education.

If my understanding above is correct, then I suppose their main aim of the study was to reveal the incidence of kusho behaviors among elderly people in Japan. The authors might originally have no interests in the mechanism of that. However, they happened to find a significant difference in years of education, which was included in questionnaire entries to obtain the basic information of subjects. By this observation, they found themselves to be obliged to discuss that in the paper.

Am I right?

If it was the case, I would like authors to more clearly indicate these backgrounds. Otherwise, I feel a difficulty to get their points in the current version of the manuscript.

Also, I would like authors to more explicitly explain why they considered it was important to reveal the incidence of kusho among the elderly in Japan. I suppose this was the main claim that authors should have appealed.

Actually, I think it might be better if they had an interest in the investigation of the mechanism of kusho from the beginning. They might want to measure “sensorimotor system” (line 47) possibly using RT of a button press to a visual cue and “cognitive functions” (lines 47-48) using assessments. They tried to use 100 RAKAN (line 95) for the latter, but unfortunately, it was not completed to all participants.

Here, for the next chance of the continuation of this study, another idea would be Japanese Adult Reading Test (JART), which was standardized and relatively easy to conduct.

Their way of description:

Another reason that I feel a difficulty in grasping their points is their way of writing the manuscript. I will raise two points: paragraph writing (again) and reduction of sentences beginning with a particle.

(1) Paragraph writing

As I wrote in the previous review, the authors frequently put their conclusion (main statement) at the last part of the paragraph. This is a feature that is frequently observed among people from Asia. I would suggest that they can move the last sentence to the top of the paragraph.

Example #1: page 3, the paragraph beginning with “To investigate…” (line 49)

They may want to start the paragraph with the last sentence of this paragraph, i.e., “In the present study…” (lines 58 - 61). They may want to modify it such as:

We hypothesized that the elderly people would show a better cognitive performance in a kusho condition than in the other conditions because… something like this.

Example #2: page 10, the paragraph beginning with “While previous studies…” (line 266)

They may want to start with the last sentence of this paragraph, i.e., “The present results suggest that the interaction between…” (Lines 277 – 279).

Then, they could deepen and strengthen the discussion.

Example #3: page 12, the paragraph beginning with “Although this study did not clarify…” (line 324)

They may want to start with the last sentence of this paragraph, i.e., “Our data suggest that decline in a specific function may be involved with a modulation of the kūsho effect in the kanji construction task” (lines 343-344).

(2) Reduction of sentences beginning with a particle

There are too many sentences beginning with a particle (e.g., For…, In the…) . Please reduce them by half.

By the way, did authors ask a copy-editing company? If not, they should ask.

Minor points.

(1) Please remove discussion from Materials and Methods.

(2) Line 153. What did they mean by “only one visual condition”? My understanding is that authors had 3 conditions, as shown in Figure 2. In lines 349-350, they also wrote; “this study used only one visual condition instead of comparing the two visual conditions.” Please clarify what was “one” or “two”.

(3) Authors frequently use the word “hand condition”. Could it be just “experimental condition,” because there were no “foot” conditions?

(4) I found a discrepancy in the description of RT.

At line 121, they wrote; “Response times were not statistically analyzed or discussed because participants’ voices were not always loud or clear enough to detect the onset of the responses, and the ratio of missing values was very high (31% of correct responses were missing values).”

However, they reported RT in line 209; “The average response times were 6.9 s (SD = 1.2) in the kūsho condition, 6.5 s (SD = 1.8) in the static condition, and 6.8 s (SD = 1.3) in the circle-drawing condition.”

Did they mean that the report was for the remaining 69% (i.e., 100 – 31) of responses?

Also, how did they measure RT?

Please clarify.

7. PLOS authors have the option to publish the peer review history of their article (what does this mean?). If published, this will include your full peer review and any attached files.

Reviewer #1: No

Reviewer #2: Yes: Claudia Rodríguez-Aranda

Reviewer #3: No

---

## [Author Response · Author response to Decision Letter 1]

2 Oct 2019

Dear Editor and Reviewers,

We greatly appreciate your rigorous and fair review of the manuscript as well as your insightful comments. The revised manuscript has been modified in response to the comments, as follows:

All comments from the reviewers have been responded to in the following pages. In the revised manuscript, revisions are highlighted using violet font (red is for the first revision). 

We look forward to hearing from you regarding our submission. We would be glad to respond to any further questions and comments that you may have.

Sincerely,

Yoshihiro Itaguchi

---

## [Decision Letter · Decision Letter 2]

27 Nov 2019

PONE-D-19-16768R2

Writing in the air: facilitative effects of finger writing in older adults

PLOS ONE

Dear Dr. Itaguchi,

Thank you for submitting your manuscript to PLOS ONE. After careful consideration, we feel that it has merit but does not fully meet PLOS ONE’s publication criteria as it currently stands. Therefore, we invite you to submit a revised version of the manuscript that systematically addresses all the points raised during the review process. Please see my comments below, and pay particular attention to the additional revisions indicated by Reviewers 1 and 3.

We would appreciate receiving your revised manuscript by Jan 11 2020 11:59PM. To enhance the reproducibility of your results, we recommend that if applicable you deposit your laboratory protocols in protocols.io, where a protocol can be assigned its own identifier (DOI) such that it can be cited independently in the future. For instructions see: http://journals.plos.org/plosone/s/submission-guidelines#loc-laboratory-protocols

We look forward to receiving your revised manuscript.

Kind regards,

Emmanuel Manalo, PhD

Academic Editor

PLOS ONE

Additional Editor Comments (if provided):

This second revision is much improved, and the reviewers' comments reflect that improvement in the quality of the manuscript.

However, please note that both Reviewers 1 and 3 have indicated further (more minor) revisions that need to be made, so can you please carefully and systematically address all of those? Note that Reviewer 1 included an annotated version of your manuscript showing some of the revisions that need to be made.

If you can please provide your responses on a point-by-point basis, it would make it easier for me to make the next decision on your manuscript. Depending on the extent to which you have clearly implemented the modifications indicated by the reviewers, it may not be necessary for me to send your next revision to them for further review.

Reviewers' comments:

Reviewer's Responses to Questions

**Comments to the Author**

1. If the authors have adequately addressed your comments raised in a previous round of review and you feel that this manuscript is now acceptable for publication, you may indicate that here to bypass the “Comments to the Author” section, enter your conflict of interest statement in the “Confidential to Editor” section, and submit your "Accept" recommendation.

Reviewer #1: All comments have been addressed

Reviewer #2: All comments have been addressed

Reviewer #3: (No Response)

2. Is the manuscript technically sound, and do the data support the conclusions?

Reviewer #1: Partly

Reviewer #2: Yes

Reviewer #3: Yes

3. Has the statistical analysis been performed appropriately and rigorously? 

Reviewer #1: I Don't Know

Reviewer #2: Yes

Reviewer #3: Yes

4. Have the authors made all data underlying the findings in their manuscript fully available?

Reviewer #1: Yes

Reviewer #2: Yes

Reviewer #3: Yes

5. Is the manuscript presented in an intelligible fashion and written in standard English?

Reviewer #1: No

Reviewer #2: Yes

Reviewer #3: Yes

6. Review Comments to the Author

Reviewer #1: I feel that the ms. is improved, and although the research behind it still not ideally designed, it is publishable. In my attached text I make several suggestions for improving the diction and clarity of the ms., and I would strongly suggest that the authors consider these several small points.

Reviewer #2: The manuscript has been greatly improved and the authors have addressed all my comments.

I have no additional concerns or suggestions.

Reviewer #3: I found that the manuscript has been greatly improved. In the previous versions of the manuscript, their description and discussion were quite superficial. Specifically, it previously failed to sufficiently discuss the underlying cognitive processes of kusho behaviors, which did not qualify the appearance in the journal; they just appeared to conduct the experiment by bothering elderly people and displayed the results without a sufficient discussion. I think their biggest fault would be their lack of a specific hypothesis regarding the “facilitative effect” of kusho; in short, facilitate what? Why can the kusho improve the ratio of correct answers in the kanji construction puzzle? According to their Introduction, they appear to have no ideas for that. (I would say, it appears they cannot understand what reviewers really suggested in this point.)

However, their discussion has been sufficiently deepened and well described after these review processes. Actually, I would say they are still a little superficial and have not fully discussed some important points that this topic (kusho) has raised in the research field so far, specifically, the underlying cognitive procedures including aspects in the visual feedback. Nonetheless, considering the limitation of the publication, I think they adequately mentioned the points related to the kusho observed in the elderly people in the latest version.

I hope the authors continue to investigate the mechanism of the kusho behaviors to conduct experiments that focus on “specific cognitive functions” that they argued in Discussion.

I would note a minor point.

Page 5, lines 116-118.

In Fig 1a, for example, to make the original kanji character, all subparts were vertically compressed while the top subpart was placed such that it overlapped with the other parts.

The example in the Fig. 1a does not appear to fit to the explanation; the explanation also does not match to the caption of Fig. 1.

7. PLOS authors have the option to publish the peer review history of their article (what does this mean?). If published, this will include your full peer review and any attached files.

Reviewer #1: No

Reviewer #2: Yes: Claudia Rodríguez-Aranda

Reviewer #3: No

---

## [Author Response · Author response to Decision Letter 2]

2 Dec 2019

All responses are included in the attached file.

---

## [Editor Report · Decision Letter 3]

9 Dec 2019

Writing in the air: facilitative effects of finger writing in older adults

PONE-D-19-16768R3

Dear Dr. Itaguchi,

We are pleased to inform you that your manuscript has been judged scientifically suitable for publication and will be formally accepted for publication once it complies with all outstanding technical requirements.

With kind regards,

Emmanuel Manalo, PhD

Academic Editor

PLOS ONE
---

## [Editor Report · Acceptance letter]

11 Dec 2019

PONE-D-19-16768R3 

Writing in the air: facilitative effects of finger writing in older adults 

Dear Dr. Itaguchi:

I am pleased to inform you that your manuscript has been deemed suitable for publication in PLOS ONE. Congratulations! Your manuscript is now with our production department. 

With kind regards,

on behalf of

Professor Emmanuel Manalo 

Academic Editor

PLOS ONE